# Polygenic and socioeconomic risk for high body mass index: 69 years of follow-up across life

David Bann[1]*, Liam Wright[1]*, Rebecca Hardy[2,3], Dylan M. Williams[4,5], Neil M. Davies[6,7,8]

1 Centre for Longitudinal Studies, Social Research Institute, UCL, London, United Kingdom, 2 School of Sport, Exercise and Health Sciences, Loughborough University, Loughborough, United Kingdom, 3 Social Research Institute, UCL, London, United Kingdom, 4 MRC Unit for Lifelong Health and Ageing at UCL, London, United Kingdom, 5 Department of Medical Epidemiology and Biostatistics, Karolinska Institutet, Stockholm, Sweden, 6 MRC Integrative Epidemiology Unit, University of Bristol, Bristol, United Kingdom, 7 Population Health Sciences, Bristol Medical School, University of Bristol, Bristol, United Kingdom, 8 K.G. Jebsen Center for Genetic Epidemiology, Department of Public Health and Nursing, NTNU, Norwegian University of Science and Technology, Trondheim, Norway

* david.bann@ucl.ac.uk (DB); liam.wright@ucl.ac.uk (LW)

**Data Availability Statement:** Data are available from application from the following: https://www.nshd.mrc.ac.uk/data. The code used to run the analysis is available at https://osf.io/psbvy/.

**Funding:** DB is supported by the Economic and Social Research Council (grant number ES/

## Abstract

Genetic influences on body mass index (BMI) appear to markedly differ across life, yet existing research is equivocal and limited by a paucity of life course data. We thus used a birth cohort study to investigate differences in association and explained variance in polygenic risk for high BMI across infancy to old age (2–69 years). A secondary aim was to investigate how the association between BMI and a key purported environmental determinant (childhood socioeconomic position) differed across life, and whether this operated independently and/or multiplicatively of genetic influences. Data were from up to 2677 participants in the MRC National Survey of Health and Development, with measured BMI at 12 timepoints from 2–69 years. We used multiple polygenic indices from GWAS of adult and childhood BMI, and investigated their associations with BMI at each age. For polygenic liability to higher adult BMI, the trajectories of effect size ($\beta$) and explained variance ($R^2$) diverged: explained variance peaked in early adulthood and plateaued thereafter, while absolute effect sizes increased throughout adulthood. For polygenic liability to higher childhood BMI, explained variance was largest in adolescence and early adulthood; effect sizes were marginally smaller in absolute terms from adolescence to adulthood. All polygenic indices were related to higher variation in BMI; quantile regression analyses showed that effect sizes were sizably larger at the upper end of the BMI distribution. Socioeconomic and polygenic risk for higher BMI across life appear to operate additively; we found little evidence of interaction. Our findings highlight the likely independent influences of polygenic and socioeconomic factors on BMI across life. Despite sizable associations, the BMI variance explained by each plateaued or declined across adulthood while BMI variance itself increased. This is suggestive of the increasing importance of chance ('non-shared') environmental influences on BMI across life.

M001660/1); DB and LW by the Medical Research Council (MR/V002147/1). DMW is funded by the UK's Medical Research Council (MC_UU_00019/2). NMD works in a unit that receives support from the University of Bristol and the UK Medical Research Council (MC_UU_00011/1) and is supported by a Norwegian Research Council Grant number 295989. The funders had no role in study design, data collection and analysis, decision to publish, or preparation of the manuscript.

**Competing interests:** The authors have declared that no competing interests exist.

## Author summary

We sought to better understand how polygenic and socioeconomic risk for high body mass index (BMI) differed across life, using data from a birth cohort followed-up from 2 to 69 years. High polygenic risk for adult BMI was associated with greater absolute differences in BMI at older ages, yet the explained variance peaked in early adulthood and plateaued thereafter. For polygenic risk for high childhood BMI, explained variance was largest in adolescence and early adulthood; effect sizes were marginally smaller from adolescence to adulthood. Low socioeconomic position was also associated with high BMI—effect sizes increased across life yet explained variance plateaued across adulthood. The discrepancy between effect sizes and explained variance was likely due to the phenotypic variance in BMI increasing across life: the increase in BMI variance matched or exceeded the increase in effect sizes. Inasmuch as our study captured key genetic and shared environmental influences on BMI, our findings suggest that chance ('non-shared') environmental influences may be increasingly important for BMI at later ages. Finally, we found little evidence for interactions between socioeconomic position and polygenic indices; rather, both were independently associated with BMI. Our findings thus highlight the importance of both environmental and genetic factors for BMI across life.

## Introduction

Body mass index (BMI) is an important modifiable determinant of population health—its prevalence markedly increased from the 1980s onwards, and remains persistently high [1, 2]. This drastic increase demonstrates the importance of environmental influences on BMI—population genetics do not change over such a short time span. Continuing evidence, however, has emerged on the link between genetic propensities and the level of BMI. For example, twin study estimates of heritability of BMI range from 47% to 90% [3]–with heritability typically highest in childhood. Polygenic indices in unrelated individuals predict approximately 8.5% of the variance in BMI [4, 5].

Better understanding changes across life in the genetic determinants of BMI may inform etiology, the timing of preventative or weight loss efforts, and the interpretation of increasing number of studies utilizing genetically-informed designs to study BMI as either an exposure or outcome of interest [6–8]. Studies investigating genetic variation in the gene *FTO*—the first variant reliably linked to higher BMI—have repeatedly found that effect sizes are largest in earlier adulthood [9]. However, BMI is a complex and polygenic trait [4, 7, 10], necessitating a need to investigate how polygenic predictors of BMI differ across life.

Recent studies have used polygenic indices (also referred to as polygenic scores) to investigate associations with BMI at different life stages [6]. However, interpretation is hampered by a paucity of data across life on the same individuals. While samples of multiple birth cohorts can be used to approximate how associations differ by age, they may be confounded by the sizable cohort differences in links between polygenic indices and BMI [11]. Further, multiple polygenic indices now exist for both childhood and adulthood BMI; assessment of these across a large age span in a single cohort would aid interpretation of their use. Indeed, it has been argued that the genetic underpinning of obesity is relatively constant across life [10][4, 7].

Other gaps in evidence motivate the need for future work. First, increases in BMI across life are marked by increases in its mean and its variance, and corresponding increases in BMI at upper values (above overweight and obesity thresholds) [12]. Conventional analytical approaches such as linear regression solely investigate mean differences. There is suggestive

evidence that the influence of genetic factors is strongest amongst those already higher in weight where health risks are greatest [13, 14], yet this requires replication and formal testing.

Second, it is unclear how genetic and socioeconomic position (SEP) [15–17] influences on BMI operate together. It has been suggested that there may be multiplicative effects [18–20], such that genetic influences are largest amongst those in disadvantaged SEP whom have fewer resources available to protect against weight gain or to initiate/maintain weight loss [18, 19]. While this is a compelling narrative, evidence for this suggestion is equivocal, with some studies reporting weak [19] or mixed [18] results. Further, lack of replication in early gene x environment interaction studies [21] suggests that publication bias could have occurred.

Large SEP differences in BMI exist in high income countries [15–17]—those with more disadvantaged SEP typically have higher BMI, with the strength of this association widening across childhood, adolescence and adulthood [15, 16] [22]. There is also evidence that such links are causal in nature [20, 23–27], although this is not universally found [28–30]. If these associations are indeed causal, they would be anticipated to operate independently of genetic influences on BMI (e.g., be evident before and after mutual adjustment). Some [31] [32] but not all [33] recent studies which incorporate polygenic scores in their analyses provide evidence for this, while a pooled analysis of 45 twin cohorts suggested that shared environmental factors (such as SEP) may have little-to-no influence on adolescent BMI [34]. SEP is a multidimensional construct, yet previous studies typically examine a limited number of indicators of SEP. As such, it remains unclear whether SEP influences BMI across life independently of genetic influences, whether SEP and genetic effects are multiplicative, or how such processes may change across life or by SEP indicator.

We sought to address the above gaps in evidence using life course BMI data from a single national birth cohort study—this study, initiated in 1946, contains BMI data from infancy to old age. We used multiple polygenic indices, thought to indicate liability for either childhood or adult BMI. We investigated change across age in effect size and explained variance since each is likely to be informative; we also investigated the additive/multiplicative role of childhood SEP and polygenic indices for BMI, and used quantile regression analysis to investigate associations across the BMI distribution.

## Methods

### Ethics statement

The study has received ethical approval from the North Thames Multicentre Research Ethics Committee (reference 98/2/121 and 07/H1008/168) and written informed consent was provided.

### Participants

The MRC National Survey of Health and Development [NSHD] (also known as the 1946 British birth cohort) is a longitudinal birth cohort study comprised of 5362 singleton births in mainland Britain born in a single week during March 1946 [35]. The cohort has been followed-up repeatedly across life, with blood samples obtained at 53 years and subsequently used for genotyping of common genome-wide genetic variation. As previously described [36], 2989 participants were interviewed and examined in their own homes by trained research nurses at age 53; we restricted our analyses to those with valid DNA and BMI data at each age. Those interviewed were considered broadly representative of the native born population of that age [37], yet loss to follow-up has been most frequent amongst those from disadvantaged SEP groups and those in worse health [38]. As with other cohorts, mortality rates are also highest in disadvantaged SEP groups [39]. DNA was extracted from whole blood samples and

purified using the Puregene DNA Isolation Kit (Flowgen, Leicestershire, UK) according to the manufacturer's protocol.

## Measures

**BMI.** BMI (kg/m$^2$) was derived from weight and height at 12 timepoints from 2–69 years of age (see Fig 1 for all ages); these were measured by health visitors, doctors, or nurses at all ages except 20 and 26 years where only self-reported data were available.

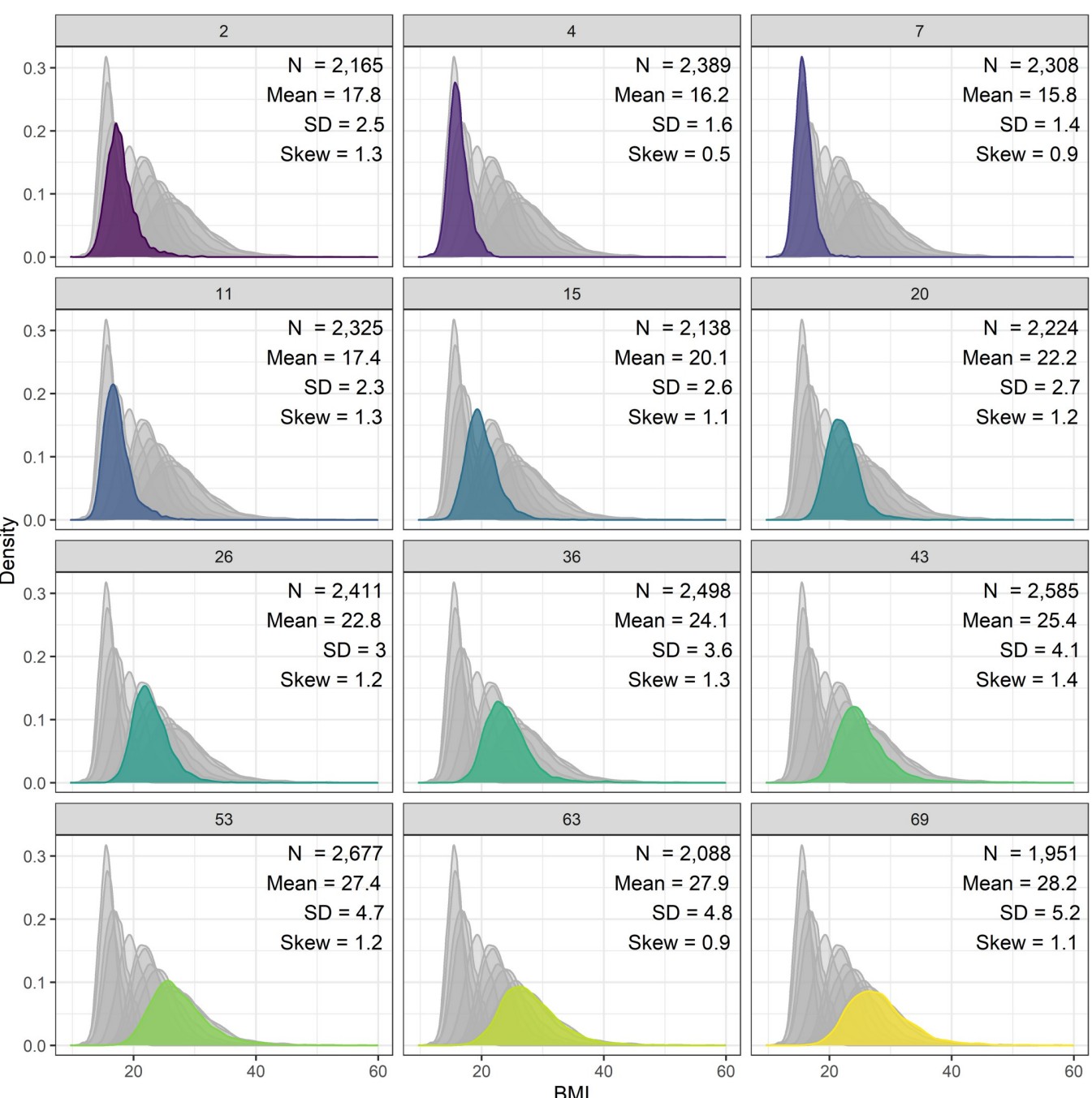

**Fig 1. Histograms of body mass index from infancy to old age in the 1946 British birth cohort sample.**

## Polygenic indices

A total of 2851 individuals were genotyped using the DrugDev microarray (assaying 476,728 SNPs) platform. Quality control (QC) analyses were performed using PLINK 1.9 [40]. Sample QC removed data on individuals with call rates < 95%, extreme heterozygosity (μ ± 3 standard deviations), sex mismatches, relatedness and duplicates ($\hat{\pi} > 2$), and principal component analysis (PCA) outliers. All participants were of European ancestry. Genotyped SNPs were excluded on the basis of the following parameters: call rate < 95%, MAF < 0.01 or HWE P < 1e-4. The genotyped SNPs were used to impute information on missing common variants using the Haplotype Reference Consortium v1.1 reference panel, accessed via the Michigan Imputation Server [41]. QC of imputed data led to SNPs being excluded with INFO score < 0.3 and MAF < 0.005. Only biallelic SNPs were retained. Following these steps, data for 2794 individuals and 8,755,070 variants were retained for polygenic score calculations. We extracted the first 10 principal components based on an LD-pruned (10kb windows; $r^2 = 0.2$) and MAF-filtered (< 0.005) variants set in approximately unrelated individuals (KING relatedness cut-off = 0.177), and projected these onto genotypes from samples in the 1000 Genomes project, phase 3 release to examine for potential sample outliers (no samples required exclusion).

In main analyses, we used a polygenic index derived by Khera et al (2019) [4] for measured BMI in adulthood which was subsequently found to strongly predict childhood BMI in independent samples and adiposity-related health outcomes. It was, at the time of writing, the polygenic index with the highest predictive capacity. This was derived using the largest GWAS available at the time [42], validated in UK Biobank data (N = 119,951), and then tested in 4 independent cohorts from birth to adulthood; N = 306,135 [4]. This yielded a score composed of 2,100,302 SNPs (not restricted to those with genome-wide significant BMI associations and including variants in linkage disequilibrium; LD).

In addition to the above, we compared analyses with other polygenic indices for high BMI. These provide alternative exposures to test how polygenic scores relate to measured BMI across life; each has relative strengths and weaknesses, and since there is no consensus on which score is most suitable for use, a comparison of findings across these indices is warranted. First, we used an alternative score for adult BMI derived from adults in UK Biobank (N = 453,169) by Richardson et al (2020) [7]. This used independent, genome-wide significant hits only for measured adult BMI (557 SNPs)—it thus provided a less predictive polygenic index, but aided comparison with other indices for childhood BMI which were also derived using significant hits only. Second, we used a score for directly measured childhood BMI derived from a meta-analysis of 41 childhood GWAS samples between ages 2 and 10 (N = 61,111) from Vogelezang et al (2020) [8]. This yielded 25 SNPs (significant independent hits only). Third, we used an additional index for high childhood BMI also based on Richardson et al (2020)[7]: adults in UK Biobank were asked to retrospectively report their weight at age 10 in categories of "thinner, plumper, or about average", yielding 295 SNPs (significant independent hits only).

SNP data and weights for the adult BMI score and retrospective childhood body size score were downloaded from the PGS Catalog using trait PGS IDs PGS000027, PGS000716 and PGS000717, respectively [5]. Equivalent data for the direct childhood scores were extracted from the relevant publication and reformatted manually. To avoid strand ambiguities in each score, we removed palindromic SNPs from the two childhood scores (palindromic SNPs were already excluded by the authors of the adulthood BMI polygenic score prior to derivation). The three scores were then calculated from NSHD genotypes using Plink 2.0 [43], assuming additive genetic effects. Scores were based on 2,083,940 SNPs (99.2%) available from the 'wide'

adult BMI score, 410 of 557 SNPs (73.6%) from the 'significant hits' adult BMI score, 234 SNPs (79.3%) of the retrospective childhood body size score, and 21 SNPs (84.0%) of the measured childhood BMI score.

## Socioeconomic position

We used childhood SEP in main analyses to reduce bias due to reverse causality, since later life BMI may affect subsequent SEP [44]. Parental social class was measured using paternal occupational class at 4 years (Registrar General's classification [RGSC]–I professional, II intermediate, III skilled non-manual, III skilled manual, IV semi-skilled, and V unskilled). This schema was used as the primary measure of social class in the UK during the period of investigation, [45] and has previously been strongly related to BMI across life [16] in this cohort. To minimize missing data, we used information at 11 years for individuals missing SEP at age 4 (n = 125). We used six other measures of SEP previously found to be related to adiposity in this cohort [46]: maternal and paternal age at leaving education at age 6 (years), maternal and paternal highest qualification (primary to secondary or higher), own highest qualification at age 26 (none to degree or higher), and own social class at age 53 (RGSC).

## Statistical analysis

First, we estimated the association of each polygenic index with BMI separately at each age (adjusted for sex) using linear regression. From these regressions, we extracted coefficients and incremental $R^2$ values to examine the size of the association and the variance explained by age. We investigated associations on the absolute ($kg/m^2$) and relative (percentage, percentile rank and standardized score) scales separately, since each may be informative; the absolute scale may aid comparability of effect sizes across adulthood as a 1 unit increase in BMI may have equivalent health risk; the relative scale may aid comparability across childhood and adulthood given sizable differences in mean BMI and its variation across age.

Second, we used quantile regression [47] to examine whether the association of BMI with polygenic indices differed across the distribution of BMI. Unlike linear regression, which estimates differences in the conditional mean of a distribution, quantile regression estimates differences in conditional quantiles of a distribution. Repeated across different quantiles, the method allowed us to examine differences in the shape (variability) and location (central tendency) of a distribution according to the values of an independent variable. We estimated quantile regressions at each decile (10th, 20th, . . ., 90th) for each polygenic index.

Third, we tested whether the relationship between SEP, polygenic indices and BMI was additive or multiplicative by regressing BMI on SEP and by including SEP × polygenic index interaction terms. We again repeated these regressions for absolute and standardized BMI indices, each polygenic index, and each measure of SEP. To simplify interpretation, we converted SEP measures in these regressions to ridit scores such that the resulting quantity in regression models shows the difference in BMI between the hypothetical top and bottom of the SEP gradient. Fourth, we examined the incremental variance in BMI explained by SEP at each age with regressions adjusting for sex, the Khera et al. index [4], and the first 10 genetic principal components. We repeated these regressions for each measure of SEP separately and also adding all measures of SEP simultaneously.

Data cleaning and analysis was conducted using R version 4.1.0 [48]. We focused interpretation on estimates and measures of precision (95% CI) rather than binary interpretation of p-values [49]. To maximise power, those with valid data for each polygenic index and BMI at each age were used in analysis with no further restrictions. Sample sizes therefore differ across analyses due to missing data for polygenic scores or BMI, loss to follow-up, death, and

emigration. All analyses using polygenic indices were adjusted for 10 principal components to help account for population stratification.

## Supplemental and sensitivity analyses

We first tested whether results differed by sex by conducting sex-stratified analyses. We then investigated if the associations were driven by weight and/or height–BMI ($kg/m^2$) is a ratio measure and thus could reflect associations with height, particularly at younger ages. To account for this, we estimated separate associations with weight, height, and BMI; we also calculated a corrected weight-for-height index, dividing weight (kg) by height raised to a power that minimized the correlation between height and the index at each age.

To explore whether focusing on complete cases at each age influenced our results, we 1) investigated whether polygenic indices were related to whether the participant had observed BMI at a given age; and 2) repeated analyses in samples with valid data for all follow-ups from a given age up to age 69, iterating across follow-ups (e.g., those followed from age 2 had complete case data at all timepoints, while those followed-up from 53 years had valid data from 53–69 years).

## Results

### Descriptive statistics

From age 7 onwards, BMI indices increased and exhibited more variability (higher SD); see Fig 1. All polygenic indices were moderately-strongly positively correlated. The Khera et al. adult index [4] was correlated with the Richardson et al. [7] (adulthood) index at r = 0.43 and at r = 0.38–0.39 with both childhood indices. The childhood indices were correlated at 0.56 with each other. There was some evidence that polygenic indices differed by childhood social class, indicating social patterning of genetic risk (S1 Fig). Notably, participants from professional backgrounds have approximately 0.2 SD lower polygenic index values (for all indices) relative to sample averages.

### Polygenic index and BMI across life

The polygenic index for BMI derived from Khera et al. [4] was positively associated with BMI at all ages; Fig 2. The size of the association was small in infancy and childhood and increased in strength from early adolescence (age 11) to older adulthood (age 69). Effects sizes were largest at age 53 and remained similarly large at ages 63 and 69. A 1 SD increase in polygenic index was associated with 1.46 (95% CI: 1.24, 1.69) $kg/m^2$ higher BMI at age 69. Findings were similar when examined in terms of percentage BMI (log transformed*100) differences (S2 Fig). However, when examined in terms of standardized BMI differences (i.e., relative to the mean and SD at each age), effect sizes remained similar from ages 15–69 (S2 Fig). This was likely due to the increasing variance of BMI across time, such that larger absolute ($kg/m^2$) effects did not equate to bigger differences relative to the sample SD. Similarly, the incremental variance ($R^2$) explained by the polygenic index peaked at age 26 and was slightly weaker thereafter—from 0.10 (95% CI: 0.08, 0.12) at age 26 to 0.08 (95% CI: 0.06, 0.10) at age 69 (Fig 2). The figures in mid-later adulthood are similar to those reported by Khera et al (~ 8.5%). [4,5]

The results of quantile regression analyses are shown in Fig 3. (Results with confidence intervals shown in S3–S6 Figs) The polygenic index and BMI associations were progressively stronger at higher BMI quantiles. These results suggest that a higher polygenic index was associated with higher variability in BMI. For example, at age 69, the association between polygenic index and BMI was over twice as large at the 90th percentile (β = 2.0; 95% CI = 1.61, 2.27) as the 10th percentile (β = 0.9; 95% CI = 0.65, 1.21).

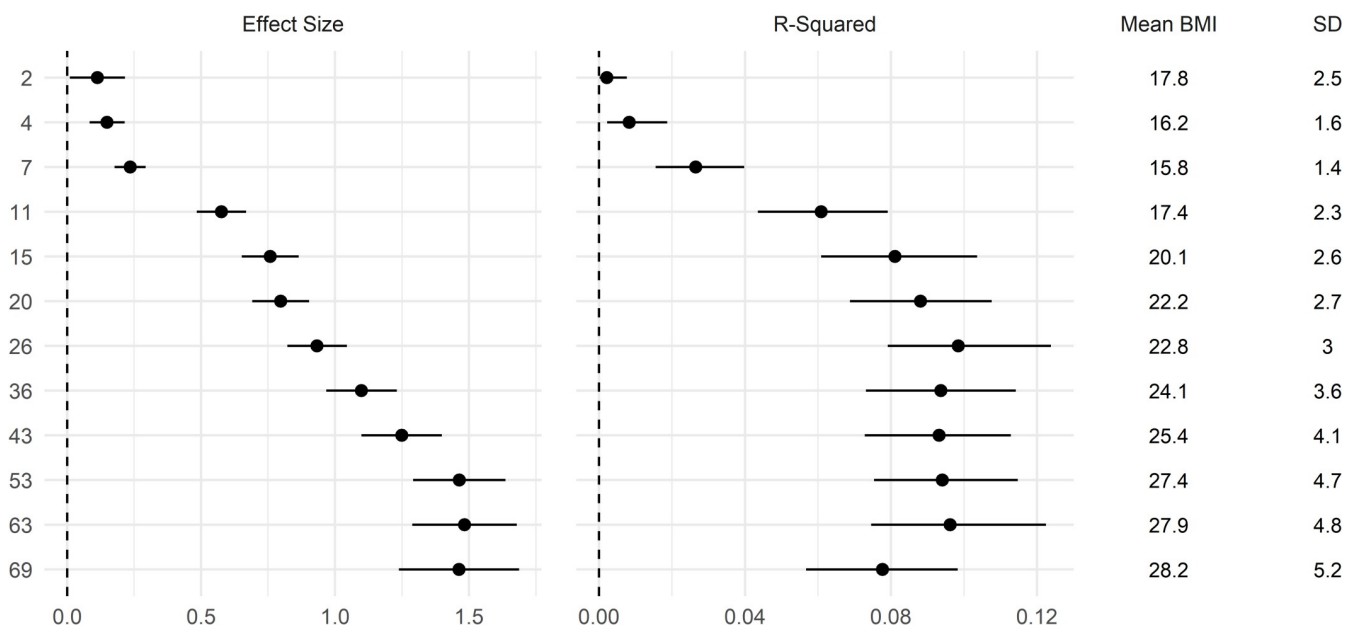

**Fig 2. Association between a polygenic index and body mass index (BMI) across life.** Drawn from OLS regressions including adjustment for sex and the first 10 genetic principal components, repeated for each polygenic index and age at follow up. Left panel: coefficients: difference in BMI per 1 SD increase in polygenic index (95% CI). Right panel: incremental $R^2$ compared with OLS regression model of BMI on sex and first 10 genetic principal components (95% CI estimated using bootstrapping [500 replications, percentile method]). Polygenic index from Khera et al [4]; used an initial sample of adults.

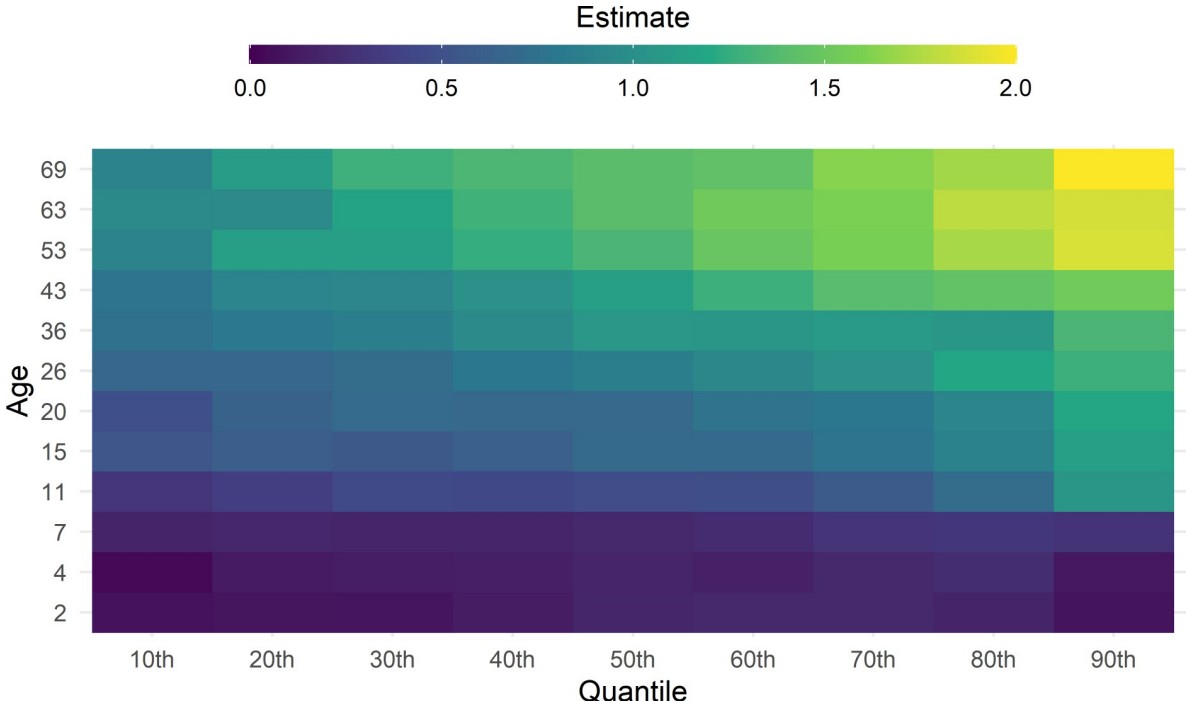

**Fig 3. Heatmap of the association between Khera et al.[4] polygenic index and BMI across life.** Drawn from quantile regressions including adjustment for sex, repeated at each follow up (y-axis) and decile (x-axis). The size of the coefficient is represented by a colour (see legend). Coefficients are interpreted analogously to linear regression: for example, Q50 shows the median (rather than mean) difference in body mass index per 1 SD increase in polygenic index.

## SEP and polygenic index in relation to BMI across life

More disadvantaged SEP in childhood (father's social class) was associated with higher BMI across multiple life stages; this association emerged from adolescence onwards—coefficients were larger at each subsequent age, and were largely unchanged after adjustment for Khera et al. [4] polygenic indices and first ten genetic principal components (top panel, Fig 4). Similar results were observed when other measures of SEP were used (S4–S7 Figs). The incremental explained variance ($R^2$) attributable to SEP was less than 3% for each indicator at each age (S8 Fig). In a model with multiple SEP indicators added simultaneously, the maximal incremental explained variance explained was approximately 4% (age 36; S8 Fig). There was little evidence of SEP (father's social class) × polygenic index interaction; coefficients for interaction terms were close to zero at all ages with confidence intervals overlapping the null in almost all cases

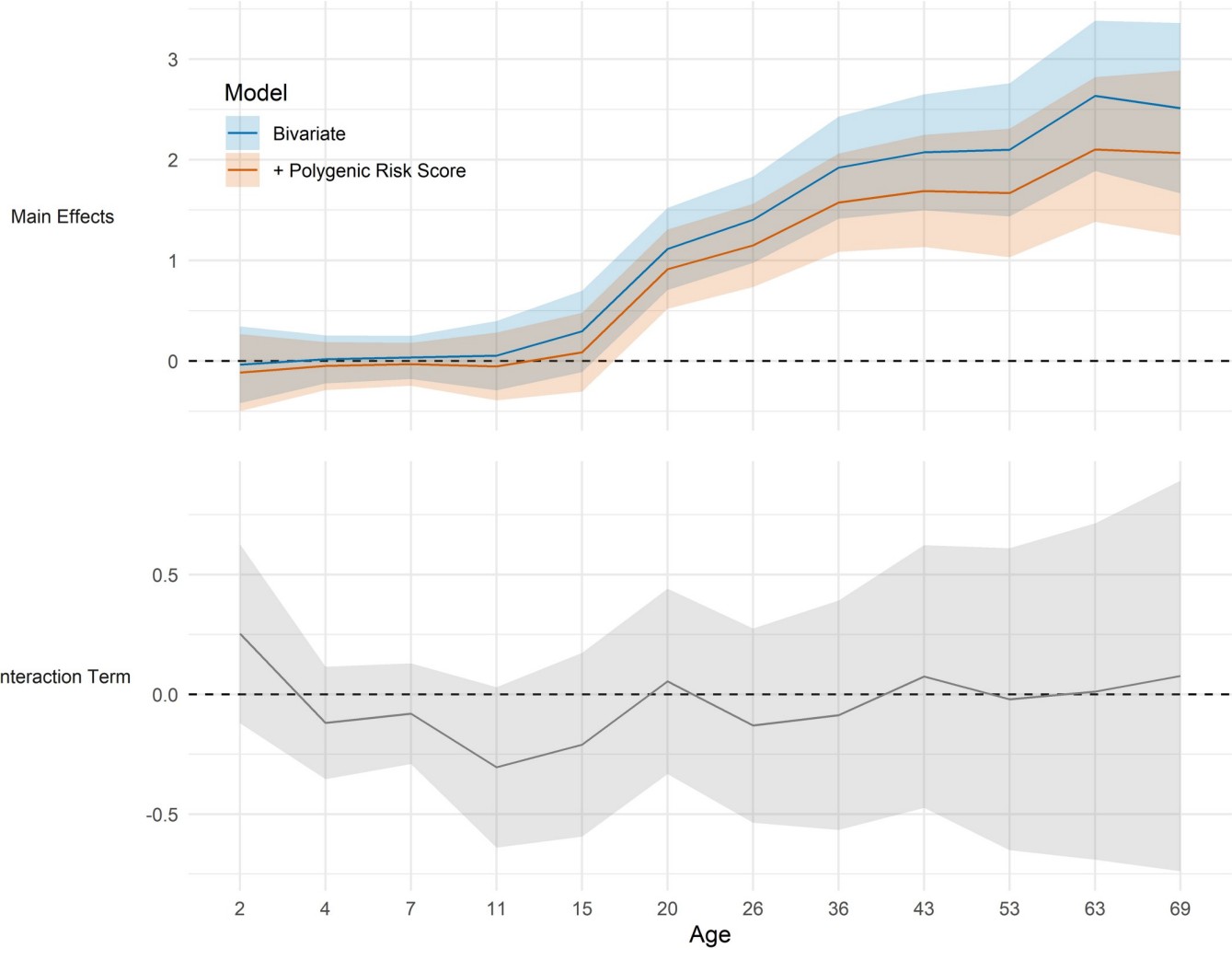

**Fig 4. Childhood socioeconomic position and polygenic index in relation to body mass index (BMI) across life.** Top panel shows the kg/m² difference in BMI in the lowest compared with highest socioeconomic position, before and after adjustment for Khera et al. (2019)[4] polygenic index for higher BMI. Bottom panel shows coefficients for the social class x polygenic index interaction term (null line is evidence for no interaction). SEP measured as father's occupational class converted to ridit score. Results from top panel drawn from OLS regression models including adjustment for sex (blue solid line) and further adjustment for polygenic indices and first ten genetic principal components (orange line). Results from bottom panel drawn from OLS regression models including adjusted for sex, polygenic index [4], first ten genetic principal components and SEP.

(bottom panel, Fig 4). Findings were similar across multiple specifications, including using different indicators of SEP (S7 Fig). Further robustness checks using standardized or log BMI yielded qualitatively similar results (data available on request).

## Comparisons of multiple polygenic indices

Polygenic indices for adulthood BMI—from Khera et al [4] (genome wide hits) and Richardson et al [7] (significant hits only) both showed larger effect sizes in later adulthood (Fig 5), though effect sizes were smaller for Richardson et al [7] and the trend of decreasing variance explained during later adulthood was more pronounced (Figs 5 and S10–S12).

Results using the Vogelezang et al. [8] polygenic index for childhood BMI are shown in Fig 5. Findings were similar when using the Richardson et al. [7] child polygenic index (S2 Fig). Effect sizes were largest in adolescence and early to mid-adulthood; associations were weak

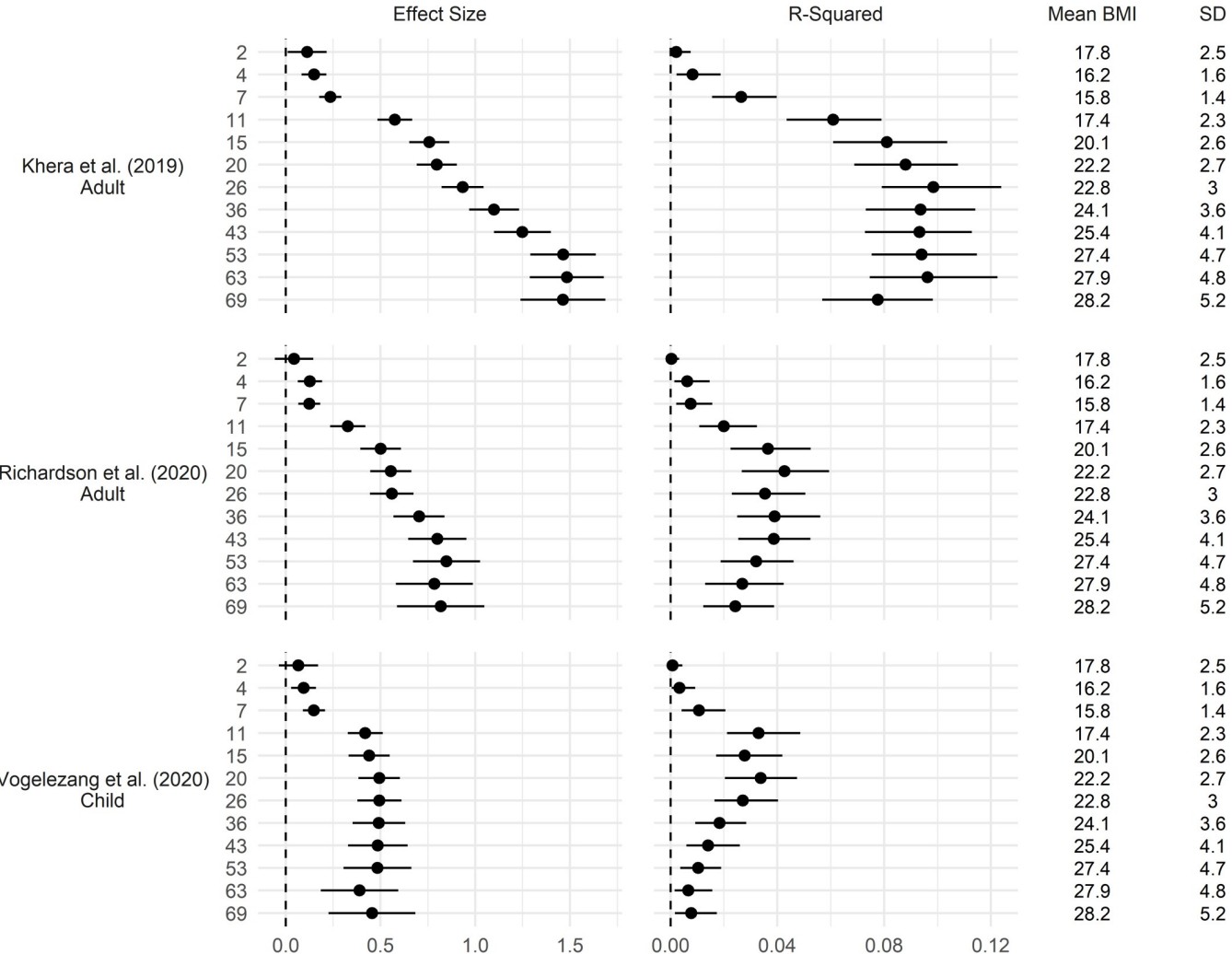

**Fig 5. Association between multiple polygenic indices and body mass index (BMI) across life.** Drawn from OLS regressions including adjustment for sex and first 10 genetic principal components, repeated for each polygenic index and age at follow up. Left panel: coefficient difference in BMI per 1 SD increase in polygenic index (+ 95% CI). Right panel: incremental $R^2$ compared to OLS regression model of BMI on sex and first 10 genetic principal components (95% CI estimated using bootstrapping [500 replications, percentile method]). Khera et al [4]: 2,100,302 SNPs (genome-wide SNPs); Richardson et al [7]: 557 SNPs (significant hits only); Vogelezang et al. [8]: 25 SNPs (significant hits only).

from ages 2–7, increased in size at age 11, and were marginally smaller at later ages (Fig 5). When examined on the relative scale (as percentage or z-score differences in BMI), the peak in effect size was more clearly evident from 11–20 (S2 Fig), with declines in association thereafter corresponding to the increased sample BMI mean and SD with age. Similarly, explained variance was highest in later adolescence to early adulthood (ages 11–20), and declined in mid to later adulthood (Fig 5). Overall, these polygenic indices explained less than 6% of variance in BMI at any age.

As with polygenic indices for high adult BMI, associations with polygenic indices for high childhood BMI were progressively stronger at higher quantiles, particularly in adolescence and young-to-middle adulthood (S4–S6 Figs); and there was little evidence for interaction with SEP (S9 Fig).

## Supplemental and sensitivity analyses

Patterns of age difference in association between polygenic indices and BMI were similar in each sex (S11 Fig), with some evidence that the associations were larger among females. Supplementary results suggested that differences in association across life were largely due to differences in weight rather than height (S13 Fig); results were also similar when using weight-for-height indices constructed at each age using an optimal power of height, to remove the association between BMI and height (S12 Fig).

Polygenic indices were related to missing BMI data at some ages (S14 Fig). Notably, non-missing BMI data at ages 63 and 69 was related to lower than average Khera et al. [4] polygenic indices. Investigation of associations between polygenic indices and BMI using samples of the same participants across time showed broadly similar results as the main analysis (S15–S22 Figs). However, there was some evidence that the plateauing of effect sizes in the Khera et al. [4] and Richardson et al. [7] (adult) index-BMI associations from 53 to 69 years was an artefact of differences in the samples at each age—when using the same sample across this age span, effect sizes were slightly higher at ages 63 and 69 (S15 and S16 Figs).

## Discussion

### Summary of findings

Using life course BMI data spanning 2–69 years of age, and multiple polygenic indices for higher childhood and adulthood BMI, we found:

1. For polygenic liability to high adult BMI (Khera et al. [4]), the trajectories of effect size and explained variance diverged across life: explained variance peaked in early adulthood and plateaued thereafter, while absolute effect sizes increased throughout adulthood.

2. For polygenic liability to high childhood BMI, explained variance was largest in adolescence and early adulthood; effect sizes were marginally smaller from adolescence to adulthood.

3. All polygenic indices were related to higher variation in BMI; effect estimates were sizable and larger at the upper end of the BMI distribution.

4. Childhood socioeconomic and polygenic risk for higher BMI across life appear to operate additively; with little evidence of interactions. The explained variance attributable to SEP on BMI was similar across adulthood.

## Comparison with previous studies and explanation of findings

Our findings are consistent with recent studies which used polygenic indices for higher adult BMI, typically in separate cohorts of different age spans. For example, Khera et al. [4] reported increasing effect sizes in regional cohorts followed-up from infancy to early adulthood (18 or 25 years). Sanz-de-Galdeano et al. [50] used separate cohorts and reported increasing strength of association from adolescence to early adulthood; while findings from another cohort suggested stability of effect sizes across older age. The results here show that these findings may generalise to a single population born in 1946 and followed across the life course (2 to 69 years). Our finding that explained variance plateaued across adulthood is consistent with results from twin studies which report declines of broad-sense heritability of BMI from adolescence across adulthood [3][51]. Further research is required to examine how these results may differ according to factors which could feasibly modify age-related changes in polygenic risk for high BMI—for instance, year of birth, ancestry, and country; the cohort used in this study was of European ancestry and was exposed to post-war rationing then an increasingly obesogenic environment across midlife [52]. In more recently born cohorts, the mean and variability of BMI is seemingly higher, yet associations with (and the predictive capacity of) polygenic indices may differ; evidence from Norway for example suggests that effect sizes for polygenic indices on BMI increased from 1960s to the 2000s. [11]

Our findings using polygenic indices for higher childhood BMI are also consistent with existing findings [6, 8]–indices derived using either recalled childhood weight or objectively measured childhood weight both have larger explained variance in childhood/adolescence/early adulthood. Our results suggest that such indices remain associated with higher BMI throughout early, mid and later adulthood; this may limit the power of multivariable Mendelian randomization studies using childhood indices of BMI in relation to later outcomes.

Differences in findings across polygenic indices suggests there may be age-specific effects via the same pathway, or multiple pathways which link genetic liability to higher BMI, despite positive correlations between the indices (likely due to some combination of overlapping SNP coverage and linkage disequilibrium). For example, genetic variants which have particularly stronger influence in early life may capture accelerated tempo of early life growth [53]. Multiple studies investigating *FTO* have reported largest effect sizes in early adulthood [9]: this was also found for the cohort used in this paper for *FTO* [36] and its nearby variants [53]. Further work is therefore required to elucidate the biological and behavioural mechanisms which link these polygenic indices to higher BMI.

Our findings highlight the importance of environmental influences on BMI across life. First, more disadvantaged childhood SEP was associated with higher adult BMI independently of polygenic risk—this was evident despite the relative crudeness of SEP measurement (e.g., a 6-category classification system for social class, compared with a polygenic index summarising information for millions of SNPs; we thus caution against the comparison of explained variances); second, the mean and variance of BMI increased across life—this may be due to environmental influences, since the explained variance attributable to polygenic indices plateaued in early adulthood. The fact that explained variance attributable to polygenic indices and childhood SEP remained similar across adulthood is suggestive of either 1) the increasing importance of chance or 'non-shared' environmental factors being increasingly important causes of between person in BMI variability across life; or 2) the increasing relative importance of other factors not measured in this study (e.g., other dimensions of SEP across life, individual behaviour independent of childhood SEP, or other genetic effects). It is notable that other traits have contrasting heritability patterns across life. For example, the heritability of cognitive performance appears to strengthen across life [54], potentially due to genetic influences indirectly

influencing future environments which in turn strengthen genetic influence. In the context of BMI, such pathways may be sizably weaker relative to the large variability in the environment which influences BMI. Finally, all polygenic indices were associated with greater variability in BMI, with effect sizes largest in higher BMI centiles—one possible cause of this is the influence of unmeasured modifiers of association which may be environmental in origin [55].

## Strengths and limitations

Strengths of this study include the use of life course data on a national birth cohort sample and use of multiple polygenic indices. Further, our analytical strategy enabled estimation of life course trajectories of effect size and explained variance; previous studies have tended to focus on either set of results, yet both are informative. Our analysis also enabled formal testing of distributional effects, and the testing of the independent and/or multiplicative role of childhood SEP. Yet the necessary use of historic data had some inherent limitations. First, the cohort preceded the wider availability of body composition measures—thus, we cannot distinguish associations of fat or lean mass across the life course. It is possible that findings with these phenotypes may differ—lean mass typically declines in older ages [56], a phenomenon which could influence associations between polygenic indices and BMI (e.g., if polygenic indices are more strongly related to fat than lean mass associations with BMI would appear to strengthen at older ages). Second, as in other prospective longitudinal studies missing data occurred; this is generally highest amongst those from lower SEP groups and those with worse health outcomes [38]. Genotyping occurred using blood samples measured at 53 years (in 1999); those with valid BMI data in early life yet no genotyping data were not included. However, we found little evidence that early life BMI was related to likelihood of having valid genetic data at 53 years (S23 Fig), though there was evidence that higher BMI during middle adulthood was related to having missing genetic data. Loss to follow-up occurred following genotyping from 53 to 69 years; our results suggest that the associations at later ages may have been downwardly biased. Finally, we provided evidence on two main dimensions of childhood SEP (parental social class and education); while these have been found to strongly predict adiposity in this cohort [16, 46], SEP is multidimensional, and it is possible that results may differ according to other dimensions such as parental disposable income, wealth, or area-level factors [57], or to SEP exposure across early and later adulthood.

## Conclusion

Our findings suggest sizable polygenic effects on BMI which differ in terms of size of association and explained variance across life. Findings also highlight the importance of the environment—adverse early life SEP was associated with higher BMI independently of polygenic risk, and increases in the population mean and variability of BMI across adulthood lead to stability of explained variance despite increasing effect sizes.

## Supporting information

**S1 STROBE Checklist. STROBE Statement—Checklist of items that should be included in reports of *cohort studies.***
(DOC)

**S1 Fig. Average polygenic indices by father's occupational class at age 4 (95% confidence intervals).**
(DOCX)

**S2 Fig. Association between polygenic indices and BMI, measured as (left to right) raw scores, standardized values, logarithms, and percentile ranks.** Standardization and percentile ranks calculated at each age at follow-up. Drawn from OLS regressions including adjustment for sex and first 10 genetic principal components, repeated for each polygenic index and age at follow up. Confidence intervals estimated using bootstrapping (500 replications, percentile method).
(DOCX)

**S3 Fig. Association between Khera et al. (2019) [4] polygenic index and (absolute) BMI.** Drawn from quantile regressions including adjustment for sex and first 10 genetic principal components, repeated at each follow up (x-axis) and decile (panels).
(DOCX)

**S4 Fig. Association between Richardson et al. (2020) [7] adult polygenic index and (absolute) BMI.** Drawn from quantile regressions including adjustment for sex and first 10 genetic principal components, repeated at each follow up (panels) and decile (x-axis).
(DOCX)

**S5 Fig. Association between Vogelezang et al. (2020) [8] [polygenic index and (absolute) BMI.** Drawn from quantile regressions including adjustment for sex and first 10 genetic principal components, repeated at each follow up (panels) and decile (x-axis).
(DOCX)

**S6 Fig. Association between Richardson et al. (2020) [7] child polygenic index and (absolute) BMI.** Drawn from quantile regressions including adjustment for sex and first 10 genetic principal components, repeated at each follow up (panels) and decile (x-axis).
(DOCX)

**S7 Fig. Association between BMI childhood socioeconomic position and SEP and body mass index (BMI) across life, by measure of SEP.** Top panel shows the $kg/m^2$ difference in BMI in the lowest compared with highest socioeconomic position. Bottom panel shows coefficients for the social class x polygenic index interaction term (null line is evidence for no interaction). Results from top panel drawn from OLS regression models including adjustment for sex (blue solid line) and further adjustment for Khera et al. (2019) [4] polygenic index and first 10 genetic principal components (orange dashed line). Results from bottom panel drawn from OLS regression models including adjusted for sex, polygenic index index (Khera et al., 2019) [4], first ten genetic principal components and SEP.
(DOCX)

**S8 Fig. Proportion of variation in BMI explained by social class.** Incremental $R^2$ compared to OLS regression model of BMI on sex and Khera et al. [4] polygenic index and first 10 genetic principal components. Multiple adjusted refers to model in which all SEP measures displayed were added simultaneously.
(DOCX)

**S9 Fig. Interaction effect between polygenic risk and childhood socioeconomic position and body mass index.** Null line is evidence for no interaction. Results drawn from OLS regression models repeated for each definition of childhood SEP (rows) and polygenic index (column) and including adjustment for sex, polygenic index score, first 10 genetic principal components, and SEP.
(DOCX)

**S10 Fig. Association between polygenic indices and body mass index (BMI).** Drawn from OLS regressions including adjustment for sex and first 10 genetic principal components, repeated for each polygenic index and age at follow up. Left panel: coefficient difference in BMI per 1 SD increase in polygenic index (95% CI). Right panel: incremental $R^2$ compared to OLS regression model of BMI on sex and first 10 genetic principal components (95% CI estimated using bootstrapping [500 replications, percentile method]).
(DOCX)

**S11 Fig. Association between polygenic indices and BMI.** Drawn from OLS regressions including adjustment for the first 10 genetic principal componetns, repeated for each sex, polygenic index, and age at follow up. Left panel: coefficient difference in BMI per 1 SD increase in polygenic index (95% CI). Right panel: incremental $R^2$ compared to OLS regression model of BMI on sex and first 10 genetic principal components (95% CI estimated using bootstrapping [500 replications, percentile method]).
(DOCX)

**S12 Fig. Association between polygenic indices and corrected BMI, measured as (left to right) raw scores, standardized values, logarithms, and percentile ranks.** Standardization and percentile ranks calculated at each age of follow-up. Correction calculated by finding x such that correlation between corrected BMI ($kg/m^x$) and height is minimized at a given age. Drawn from OLS regressions including adjustment for sex and first 10 genetic principal components and repeated for each polygenic index and age at follow up.
(DOCX)

**S13 Fig. Correlation between polygenic indices and BMI, height and weight by follow-up.**
(DOCX)

**S14 Fig. Difference in average PRS scores (95% CI) by whether participant had observed or missing BMI scores at a given age.** Drawn from separate regressions for each combination of PRS score (columns) and age of follow-up (rows).
(DOCX)

**S15 Fig. Association between Khera et al. (2019) [4] polygenic index and (absolute) BMI.** Drawn from OLS regressions including adjustment for sex and first 10 genetic principal components and repeated at each follow up using observed sample and samples of participants interviewed at each sweep following a given age (panels).
(DOCX)

**S16 Fig. Association between Richardson et al. (2020) [7] adult polygenic index and (absolute) BMI.** Drawn from OLS regressions including adjustment for sex and first 10 genetic principal components and repeated at each follow up using observed sample and samples of participants interviewed at each sweep following a given age (panels).
(DOCX)

**S17 Fig. Association between Vogelezang et al. (2020) [8] polygenic index and (absolute) BMI.** Drawn from OLS regressions including adjustment for sex and first 10 genetic principal components and repeated at each follow up using observed sample and samples of participants interviewed at each sweep following a given age (panels).
(DOCX)

**S18 Fig. Association between Richardson et al. (2020) [7] child polygenic index and (absolute) BMI.** Drawn from OLS regressions including adjustment for sex and first 10 genetic principal components and repeated at each follow up using observed sample and samples of

participants interviewed at each sweep following a given age (panels).
(DOCX)

**S19 Fig. Incremental proportion of variance in (absolute) BMI explained by Khera et al. (2019) [4] polygenic index.** Drawn from OLS regressions compared solely adjusting for sex and first 10 principal components. Regression models repeated at each follow up using observed sample and samples of participants interviewed at each sweep following a given age (panels). Confidence intervals derived from 500 bootstrap replications using the percentile method.
(DOCX)

**S20 Fig. Incremental proportion of variance in (absolute) BMI explained by Richardson et al. (2020) [7] adult polygenic index.** Drawn from OLS regressions compared solely adjusting for sex and first 10 principal components. Regression models repeated at each follow up using observed sample and samples of participants interviewed at each sweep following a given age (panels). Confidence intervals derived from 500 bootstrap replications using the percentile method.
(DOCX)

**S21 Fig. Incremental proportion of variance in (absolute) BMI explained by Vogelezang et al. (2020) [8] polygenic index.** Drawn from OLS regressions compared solely adjusting for sex and first 10 principal components. Regression models repeated at each follow up using observed sample and samples of participants interviewed at each sweep following a given age (panels). Confidence intervals derived from 500 bootstrap replications using the percentile method.
(DOCX)

**S22 Fig. Incremental proportion of variance in (absolute) BMI explained by Richardson et al. (2020) [7] child polygenic index.** Drawn from OLS regressions compared solely adjusting for sex and first 10 principal components. Regression models repeated at each follow up using observed sample and samples of participants interviewed at each sweep following a given age (panels). Confidence intervals derived from 500 bootstrap replications using the percentile method.
(DOCX)

**S23 Fig. Association between BMI and missing polygenic index by age at follow-up (95% confidence intervals).** Drawn from bivariate linear regression models repeated at each age.
(DOCX)

## Acknowledgments

This work was undertaken amidst three separate periods of paternity leave; we (DMW, DB, LW) are indebted to our partners for their continued support. We also thank George Davey Smith for a helpful comment on an earlier draft.

## Author Contributions

**Conceptualization:** David Bann, Liam Wright, Neil M. Davies.

**Data curation:** Dylan M. Williams.

**Formal analysis:** Liam Wright, Dylan M. Williams.

**Funding acquisition:** David Bann.

**Investigation:** David Bann, Liam Wright, Rebecca Hardy, Dylan M. Williams, Neil M. Davies.

**Methodology:** David Bann, Liam Wright, Rebecca Hardy, Dylan M. Williams, Neil M. Davies.

**Project administration:** David Bann.

**Software:** Liam Wright.

**Supervision:** David Bann.

**Validation:** David Bann, Liam Wright.

**Visualization:** Liam Wright.

**Writing – original draft:** David Bann.

**Writing – review & editing:** David Bann, Liam Wright, Rebecca Hardy, Dylan M. Williams, Neil M. Davies.

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
