## [Decision Letter · Decision Letter 0]

23 Dec 2021

Dear Dr Bann,

Thank you very much for submitting your Research Article entitled 'Polygenic and socioeconomic risk for high body mass index: 69 years of follow-up across life' to PLOS Genetics.

The manuscript was fully evaluated at the editorial level and by independent peer reviewers. The reviewers appreciated the attention to an important problem, but raised substantial concerns about the current manuscript which require major reworking of the presentation to ensure the science is sound and fully presented. Based on the reviews, we will not be able to accept this version of the manuscript, but we would be willing to review a much-revised version. We cannot, of course, promise publication at that time.

If you decide to revise the manuscript for further consideration at PLOS Genetics, please aim to resubmit within the next 60 days, unless it will take extra time to address the concerns of the reviewers, in which case we would appreciate an expected resubmission date by email to plosgenetics@plos.org.

[LINK]

We are sorry that we cannot be more positive about your manuscript at this stage. Please do not hesitate to contact us if you have any concerns or questions.

Yours sincerely,

Tuomas Kilpelainen

Guest Editor

PLOS Genetics

Gregory Barsh

Editor-in-Chief

PLOS Genetics

Reviewer's Responses to Questions

**Comments to the Authors:**

Reviewer #1: Review is uploaded as an attachment.

Reviewer #2: This paper analyses several somewhat understudied topics related to the genetics of BMI using British longitudinal data. There are a many topics covered in this paper (changing associations during the life course, SES*G interactions and skewness) making the paper somewhat difficult to read. I have only a couple of comments the authors may want to consider.

Some of the references citied are a bit strange. For example, Mackenbach 2020 does not give evidence on the associations between SEP and genetic factors. Also the evidence for G*SEP (references 16-18) is somewhat mixed. Generally, I think that the references should be discussed more critically.

Interestingly, the genetic variation of BMI seems to follow closely the same age pattern as found previously in large scale twin studies (Silventoinen K et al. Am J Clin Nutr 2016 and Silventoinen K et al. Am J Clin Nutr 2017), i.e., having the peak in early adulthood and then declining. I am wondering whether it would be useful to make more systematic comparison to these twin studies to show that these two methods produce very consistent results.

Minor comments

Page 3: the first sentence: The prevalence of BMI has probably not change (mean of BMI has increased of prevalence of overweight/obesity increased).

Reviewer #3: Dear authors,

Thank you very much for the opportunity to review your research.

This manuscript reads very well and and really enjoyed reading it. However, I do not think that this is anywhere ready for submission. The data presented and the interpretation lacks robustness and in many ways looks preliminary.

For example I am surprise that such a great team does not provide the necessary descriptive of the population to allow the reader to ascertain the selection factors in the cohort they studied. Childhood SEP being an important factors of not participation, this is very important to inform (in addition to many other possible confounders or selection factors).

An other example is the lack of justification of the use of Childhood SEP in this study. Why no adult SEP indicator used?

An other example suggesting that the manuscript is still at its preliminary stage is to suggest increase BMI with age not considering that the average BMI in childhood and adulthood are not comparable (in terms of absolute value).

This is really a pity as the data look fascinating. Considering the importance of such health information and the ethical responsibility to reduce speculation in social and genetic inheritance of leaving with obesity, I can only recommend to strengthen your data by following the STROBE recommendations as instructed by the journal.

**Have all data underlying the figures and results presented in the manuscript been provided?**

Reviewer #1: Yes

Reviewer #2: Yes

Reviewer #3: **No: **Statistic analysis is only ascertain from visual interpretation of the figures

PLOS authors have the option to publish the peer review history of their article (what does this mean?). If published, this will include your full peer review and any attached files.

Reviewer #1: No

Reviewer #2: No

Reviewer #3: No

---

## [Decision Letter · Decision Letter 1]

3 May 2022

Dear Dr Bann,

We are pleased to inform you that your manuscript entitled "Polygenic and socioeconomic risk for high body mass index: 69 years of follow-up across life" has been editorially accepted for publication in PLOS Genetics. Congratulations!

Yours sincerely,

Tuomas Kilpelainen

Guest Editor

PLOS Genetics

Gregory Barsh

Editor-in-Chief

PLOS Genetics

Comments from the reviewers (if applicable):

Reviewer's Responses to Questions

**Comments to the Authors:**

Reviewer #1: The authors have responded robustly and appropriately to the reviewers' feedback.

Reviewer #2: No further comments.

Reviewer #3: Dear authors, you have done a massive job revising this manuscript.

I have no further comment. Your work has big added value for the community

**Have all data underlying the figures and results presented in the manuscript been provided?**

Reviewer #1: Yes

Reviewer #2: Yes

Reviewer #3: Yes

PLOS authors have the option to publish the peer review history of their article (what does this mean?). If published, this will include your full peer review and any attached files.

Reviewer #1: No

Reviewer #2: No

Reviewer #3: No

**Data Deposition**

http://datadryad.org/submit?journalID=pgenetics&manu=PGENETICS-D-21-01500R1

**Press Queries**

---

## [Editor Report · Acceptance letter]

22 Jun 2022

PGENETICS-D-21-01500R1 

Polygenic and socioeconomic risk for high body mass index: 69 years of follow-up across life 

Dear Dr Bann, 

We are pleased to inform you that your manuscript entitled "Polygenic and socioeconomic risk for high body mass index: 69 years of follow-up across life" has been formally accepted for publication in PLOS Genetics! Your manuscript is now with our production department and you will be notified of the publication date in due course.

With kind regards,

Agnes Pap

PLOS Genetics

On behalf of:
